# Improving Sustainability, Climate Resilience and Pandemic Preparedness in Small Islands: A Systematic Literature Review

**Stefano Moncada** [1,2,*] and **Luca Nguyen** [1]

1 Islands and Small States Institute, University of Malta, MSD 2080 Msida, Malta; luca.nguyen@um.edu.mt
2 World Health Organisation Collaboration Center on Health Systems and Policies in Small States, Islands and Small States Institute, University of Malta, MSD 2080 Msida, Malta
* Correspondence: stefano.moncada@um.edu.mt; Tel.: +356-23402117

**Abstract:** Small islands are often disproportionately impacted by external shocks, and the manner in which they build resilience is increasingly important in the face of climate change and health crises, thus impacting the attainment of their Sustainable Development Goals (SDGs). This paper discusses the results of a systematic literature review (PRISMA protocol) that set out to examine which resilience-building measures are adopted by small islands to overcome the incidence of two shocks happening simultaneously: climate change and COVID-19. This is in line with the objectives set by SDG 13 (targets 13.1/13.3) and SDG 3 (targets 3.8/3.d). While 16,369 studies fulfilled the criterion of jointly assessing pandemic, health and climate, only 662 of these mentioned small islands. Moreover, 42 studies fulfilled additional screening criteria. Within these studies, we examined whether a planetary health approach, which recognises the links between environment and health systems, was adopted. The results show that in small islands, and for such shocks in tandem, a planetary health approach is scarcely considered. However, specific actions to strengthen resilience were documented to have been effective when facing climate and health crises, which we categorised as: (i) the management of short-term risks; (ii) community actions; (iii) local government support; and (iv) long-term strategic planning.

**Keywords:** small islands; climate change; COVID-19; planetary health approach; sustainable development goals; resilience; systematic literature review

## 1. Introduction

Small islands are often disproportionately impacted by external shocks [1], especially climate change [2]. The recent public health COVID-19 crisis showed how intense the impacts can be for small islands [3,4], which are often resource constrained and located far away from key economic and political decisional centres [5]. Additionally, when such climate and health crises happen simultaneously, they further threaten the attainment of the sustainable development goals (SDGs) and impact the overall development trajectories of many small island states (SISs).

The motivation behind examining these two specific crises together stems from the understanding that while the climate change crisis has been looming for more than two decades, increasingly impacting islands, the sudden and global shock brought by the COVID-19 pandemic prompted the researchers to assess preparedness and resilience building in such combined circumstances, which are predicted to increase in numbers and intensity in the future [6], further expanding on research that focuses on permacrisis [7].

A growing body of literature has been investigating the importance of strengthening climate [8–10] and health resilience [11,12], with the aim of both showing the benefit of preparedness [13,14] but also of demonstrating the scope for interdisciplinary work [15,16]. There is a common understanding that by designing and implementing sustainable development policies in a timely manner, including attaining the SDGs, climate change and public health goals can converge to foster structural transformations [17].

There are 42 SISs that are members of the United Nations (UN), representing 23 percent of the total UN members, suggesting that large states might be the anomaly within international organisations [18]. Although the total population of all SISs does not exceed 100 million inhabitants, and although their total aggregated economic contribution to the world's GDP is relatively small, they often represent crucial tourism destinations [19,20], can be central allies in international alliances [21], and are at the forefront of tackling climate crises [22,23]. Small islands tend to exhibit characteristics that are often different from the mainland and to those of larger countries. These characteristics include, among other issues, having limited land area, a majority of socio-economic activities being located by the coast, a relatively small ration of coastal area over total landmass, the inability to exploit the full potential of economies of scale, and higher costs related to transportation and other crucial services such as waste management and environmental protection. The theories and applied research related to these issues have mostly been applied in the context of larger countries, making the study of small islands increasingly relevant, possibly bringing further evidence for enriching mainstream studies or studying islands "on their own terms" [18].

Islands, especially small island developing states (SIDSs), have often been labelled as the 'canaries in the climate change coal mine' [24] given their exposure and vulnerability to the negative impacts of climate change [25]. However, islands are also increasingly understood as proactive actors 'which the rest of the world can and should now learn from' [26].

However, how specific regions or communities may be developing such sustainable development responses to tackle climate and health crises is still largely unknown, especially in SISs. Although attention toward the challenges faced by SISs due to climate change has been increasing [27] and following the impacts of recent pandemics, research has been gaining momentum [28], lack of data and research capacity remain major problems in the context of SISs. This is especially true for island communities, which paradoxically are frequently neglected in research about climate change impacts and adaptation [29,30]. What the proposed policy solutions mean for specific island and coastal communities is still largely unknown [31,32]; how these solutions might be optimally designed and implemented is often inadequately understood, especially in communities where traditional/vernacular methods of coping or active participation at the local level are not always accounted for in the planning and implementation of sustainable development policies, nor comprehensively discussed by the academic literature.

Furthermore, the limited data available for small islands is often constructed over extensive spatial or temporal scales. While these data offer some initial understanding into potential risks, it may not be meaningful at the scale at which island communities operate. In the absence of the precise identification of local shocks, adaptation activities might not be sufficiently focused, with the risk of being inadequate or even maladaptive [33,34]. As a result, there may be a risk of delaying the undertaking of specific sustainable development interventions to enhance climate change adaptation and the attainment of SDGs, especially Good Health and Well-being (Goal 3), Sustainable Cities and Communities (Goal 11) and Climate Action (Goal 13). Although the World Health Organisation has declared the end of the COVID-19 pandemic, such sudden epidemic and pandemic events might occur more often in the future [35], also due to unsustainable production and consumption patterns [36]; therefore, expanding research in the area of risk preparedness and optimal responses is crucial to support the policy of minimising negative impacts and increasing the overall well-being of countries.

This paper presents the results of a systematic literature review aimed at examining the resilience-building measures adopted by small islands to overcome the incidence of two shocks happening simultaneously: climate change and COVID-19, focusing on holistic conceptual frameworks capable of capturing multiple crises. By identifying this specific body of knowledge, the systematic literature review also aims to map the state of knowledge of sustainable development policies to build resilience to climate change and public health crises, investigating challenges, opportunities and desirable patterns in the context of

resource-constrained places such as those of small islands, possibly shedding light on sustainable development policies that are locally meaningful; this may not only provide more suitable context specific solutions but also potentially extend existing established theories and inform policymaking in contexts often overlooked in mainstream studies, such as that of small islands.

## 2. Materials and Methods

A systematic literature review (SLR) can be described as a method for 'identifying, evaluating, and synthesizing the existing body of completed and recorded work produced by researchers, scholars, and practitioners' [37]. The SLR has four key characteristics: (i) it is systematic in the way that it follows a detailed methodological approach; (ii) it is explicit in describing the precise actions undertaken to develop it; (iii) it is comprehensive by including all applicable and most relevant material; and (iv) it is reproducible, allowing any researcher following the same approach to replicate the work [38].

Precise research questions, also central to a SLR [39], were devised while keeping in mind avoiding over-generalization and striking a balance with the key objectives set out by the research; the questions are as follows:

- What is the current state of knowledge on multidimensional crises in health and environmental dimensions applied to small islands?
- How do sustainable development policies in small islands address the simultaneous challenges of climate change and public health crises?

The 2020 PRISMA guidelines [40] were adopted to identify and select suitable peer-reviewed articles for inclusion in the SLR, acquiring material from three distinct databases, SCOPUS, Web of Science and PubMed, in order to increase the coverage of the existing body of knowledge to be retrieved [41]. We selected PubMed, Scopus, and Web of Science for our systematic literature review due to their extensive coverage and recognised credibility in hosting a comprehensive collection of peer-reviewed articles. PubMed's focus on life sciences, public health and medical literature; Scopus's broad scientific scope; and Web of Science's high-quality journal selection collectively ensured an exhaustive and diverse range of high-quality scholarly articles for our research.

The keywords or search terms related to the main objectives of the research were identified as: climate change adaptation and resilience, public health crises and small islands. This selection, as shown by Table 1, prompted the following search query:

**Table 1.** Search query strategy for the PRISMA systematic literature review.

| Database | Query | Results Yielded |
|---|---|---|
| Web of Science | ALL = (("COVID-19" OR "COVID 19" OR "pandemic" OR "health" OR "healthcare") AND ("clima*" OR "environm*" OR "planeta*") AND ("SIDS" OR "small islan*" OR "Pacific Islan*" OR "Pacific ocean" OR "Caribbean" OR "Indian ocean" OR "Atlantic ocean")) | 546 |
| Scopus | ("COVID-19" OR "COVID 19" OR "pandemic" OR "health" OR "healthcare") AND ("clima*" OR "environm*" OR "planetar*") AND ("SIDS" OR "small islan*" OR "Pacific Islan*" OR "Pacific ocean" OR "Caribbean" OR "Indian ocean" OR "Atlantic ocean") | 790 |
| PubMed | ("COVID-19" OR "COVID 19" OR "pandemic" OR "health" OR "healthcare) AND ("clima*" OR "environm*" OR "planetar*"") AND ("SIDS" OR "small islan*" OR "Pacific Islan*" OR "Pacific ocean" OR Caribbean" OR "Indian ocean" OR "Atlantic ocean") | 650 |

Publications were selected if:

- They were peer-reviewed articles or government and non-government papers, guidelines, reports or similar types exploring the topics of the review;
- They contained relevant information on environmental, climate and health crises in the context of small islands;
- If two same articles were retrieved, only the latest version was kept as the most recent;
- Published in the English language;
- Published between 2007 and March 2023.

The full list of result searches is available upon request.

### 2.1. Screening and Eligibility

After retrieving the publications from the four sources (three academic databases and Google search to capture grey literature), the information of identified items (title, abstract and authors, date, publication and DOI) were formatted in a uniform Excel file.

The items went through the following screening process. Titles and abstracts were read, and publications were excluded if:

- There were items not using solid methodologies, methods or tools. Thus, qualifying publications should require evidence of the approach used, be replicable and make a contribution to the development of the topic of the research;
- Following the above point, opinion pieces were excluded;
- The same publication was duplicated but with a different name or published in different venues, the peer-reviewed version or the most recent version was selected;
- Not relevant to small islands, SISs and/or SIDSs.

Figure 1 below shows the complete workflow of the identification of relevant studies for this systematic review.

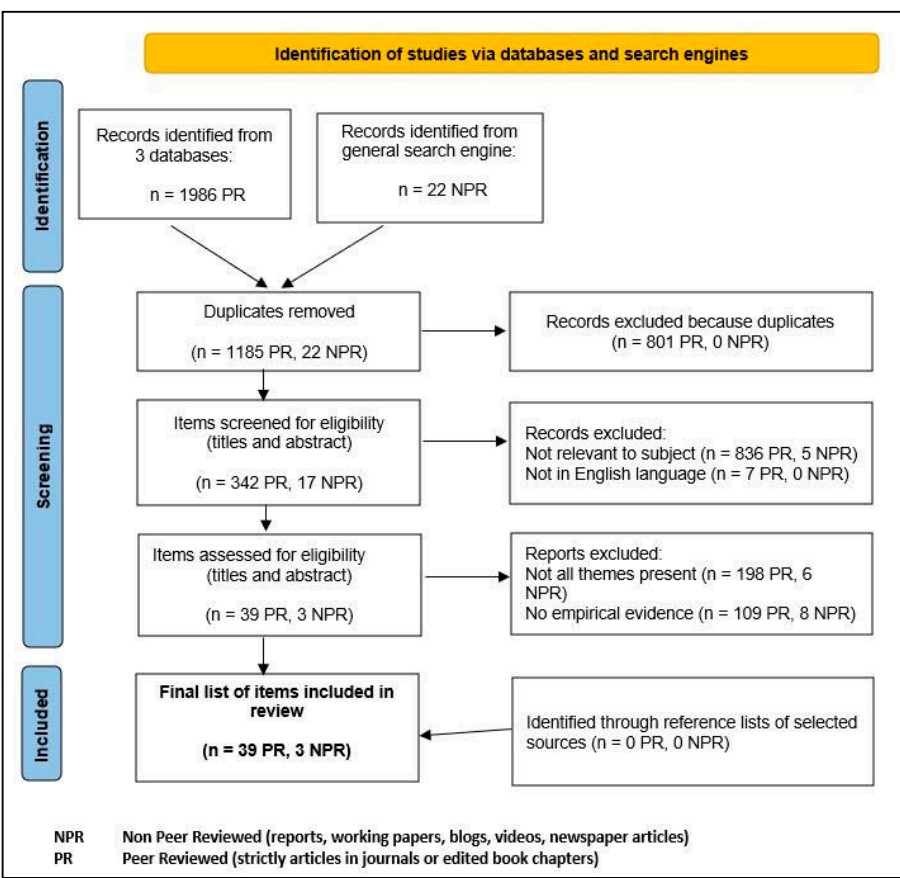

**Figure 1.** Identification of studies via databases and search engines.

*2.2. Thematic Analysis*

Building on Ryan and Bernard [42], items that made it to the final selection represented the fundamental concepts of the research, which in the case of this SLR were climate resilience and health preparedness in the context of islands. Unifying themes were then scrutinized and chosen based on the core ideas, arguments and concepts based on the main research question. The selection of the final four broad themes, building on the principles described above, were inductively originated from an all-inclusive understanding of the articles reviewed [43].

**3. Results**

*3.1. Papers Selected for the Systematic Literature Review*

A total of 2008 items were retrieved from academic peer-reviewed databases (n = 1986) and the Google Scholar search engine (n = 22), as shown by Figure 1 above below. After duplicates were excluded, 1207 unique publications remained. These items were screened, specifically with respect to whether they were in the English language and whether the items discussed the relevant topics under study in this systematic literature review. Exclusion at this stage was for instance due to:

- Articles with abstracts in English but with the main articles being in another language (n = 7 of peer-reviewed publications);
- An article discussing a vulnerability index in touristic small islands but without using empirical data to substantiate the results;
- Articles recommending policies without relevant methodologies used (n = 109 of peer-reviewed publications);
- Articles assessing general vulnerability and resilience in small islands outside the scope of the research focus (n = 198 of peer-reviewed publications).

After this stage, 42 items remained. No other items identified through other sources were added (e.g., through reference lists of the selected PR articles); thus, the final selection remained at 42 publications. The search results can be seen in Table 2 below.

**Table 2.** Selection of studies following the screening process.

| Source | Peer Reviewed | Non-Peer Reviewed |
|---|---|---|
| Academic database | 39 | 0 |
| Google Scholar | 0 | 3 |
| Other sources | 0 | 0 |

The studies identified from the SLR can be categorised according to their typology of crisis, as follows:

- 25 studies on COVID-19 post-2020;
- 8 studies on general environmental/climate crises spanning various time periods;
- 6 studies on mixed crises (pandemic + environmental/climate) from different periods;
- 1 study on influenza from the late 2010s;
- 1 study on AIDS from the 1990s;
- 1 study on other types of crises (food crisis).

*3.2. Key Themes Emerging from the SLR*

The publications that made the final selection had common themes that were identified after thoroughly reviewing them while adopting a thematic approach. The major themes that emerge from the final selection of papers can be summarised as follows:

1. Management of short-term risks;
2. Community action;
3. Local government support;
4. Long-term strategic planning.

The categories for the key themes emerging from the papers retrieved were generated from a textual analysis of the papers, consistent with the PRISMA approach for conducting systematic reviews (see Supplementary Materials). This thematic analysis was based on the frequency and prominence of concepts within the literature as well as a qualitative evaluation of each paper's primary focus, encompassing the abstract, methodology, research questions and conclusions [40].

The most mentioned theme present in 18 publications was the need for long-term strategic planning, and it accounted for 42% of items. Management of short-term risk was investigated by 28% of items (n = 12), while bottom-up community coping strategies were investigated in 31% (n = 13) of items. Lastly, 14% (n = 6) of items investigated the role of the local government during multidimensional crises in small island states. The items included in each theme are summarised in Table 3 below.

**Table 3.** Key themes identified from the systematic literature review.

| Themes | Peer Reviewed | Non-Peer Reviewed |
|---|---|---|
| Management of short-term risks | 10 | 2 |
| Community action | 13 | 0 |
| Local government support | 6 | 0 |
| Long-term strategic planning | 18 | 1 |

## 4. Discussion

An evident finding arising from the review of the selected body of knowledge is the absence of a clear comprehensive conceptual framework addressing responses to multidimensional crises in the context of islands. There still seems to be a compartmental approach to crises, often guided by the nature of the crises themselves, offering responses that are grounded in either public health or environmental management, as was the case for the COVID-19 and climate emergencies. Additionally, from the articles retrieved in the systematic literature review and analysis of the texts, the planetary health approach is underutilised in the tackling of crises in small islands.

Despite the absence of a comprehensive conceptual framework capable of addressing the preparedness and response to multiple crises, four distinct thematic patterns emerge from the reviewed literature that tend to support resilience building in islands. These are discussed in more detail below. While these patterns may not have been intentionally formulated as such, they could represent the foundation of a potential theoretical framework to categorise and explain the preparedness and responses to multiple crises in the context of islands.

### 4.1. Management of Short-Term Risks

Although many small island states often lacked a precise protocol in the management of short-term emergencies, when faced with climate and public health crises, they were able to take advantage of their insular characteristics, especially by relying on the role of networks and their inherent helicopter view [44].

This is the case for increased collaboration among different stakeholders, which resulted in a better reach to individuals and communities to test and reduce COVID-19 transmissions [45]. Similarly, in the immediate aftermath of an emergency, when foreign experts tend to depart from SIDSs (especially educational and medical services), teleworking represented an immediate relief that was generally quickly organised to cope with such limitations in the short-term while still looking for more long-term arrangements [46].

In some instances, experiences with previous public health emergencies guided SIDS to strengthen their resilience in emergency care systems, and potentially enhancing future capacity for both routine care and outbreak responses. This was the case for emergency departments in the Solomon Islands and Fiji. Responses to the Ebola outbreak in Africa guided them in restructuring their limited physical spaces, so as to create separate areas for screening, isolation, resuscitation and storage [47].

The response to climate extreme events, especially in rapidly generating and sharing after-action reports that examine shelter operations and mandated evacuations, can also guide pandemic preparedness [48]. In fact, lessons gleaned early in the season can refine and shape evacuation and sheltering protocols to develop mitigation procedures when stronger storms threaten and strike later in the season. Furthermore, by looking at how various options for evacuation and sheltering might influence the spread of pandemics, communities can manage concurrent threats of climate-driven Atlantic hurricanes superimposed on COVID-19 transmission risks [48].

Existing cooperation on climate extreme events among CARICOM countries in the Caribbean region was utilised and extended to negotiate additional resources from the international community in the face of the economic crisis triggered by COVID-19 [49].

The institutional flexibility to understand that, under multiple crises, a precise chain of command and coordination should make decisions based on resources and actual risks rather than based on a set of rigid predetermined rules on specific cases (e.g., viral pandemics, storms) resulted in many SISs creating more harmonised institutional procedures that fostered improved coordination among actors [50]. Similarly, the importance of early detection, monitoring, assessments, community engagement and awareness, reporting and surveillance, and deploying prevention strategies in responding to crises was emphasised, also highlighting the need for modernising training for environmental health inspectors to enhance their emergency preparedness competency [51].

From the literature reviewed above, it emerges that the use of flexible short-term planning by islands seems to build on the strong role of social capital, especially networking with and within institutions, as a reply to the relatively limited resource endowment that islands often exhibit. These results confirm the well-established island studies literature that identifies the resourcefulness of islanders [52] as well as governance and institutional capacity [53] as key factors for strengthening islands' resilience to adverse external shocks.

### 4.2. Community Action

The importance of community action is another central theme that emerges from the studies reviewed. The main role that stems from the involvement of community actors during crises is both oriented at filling a gap and/or supporting existing efforts by public authorities and other non-state actors in strengthening resilience in the face of external shocks.

The spontaneous organisation of selected actors within the community, with the aim of developing prevention and responding more effectively to those in need during crises, is a recurrent feature in many islands. During the peak of the spread of HIV/AIDS, senior health management in public health organisations in Fiji implemented 'independent' prevention and information programs to tackle HIV/AIDS, actively coordinating with other actors within the community [54]. These actions promoted the overall need to have a more holistic and general framework on promoting and coordinating responses from community members, which would not be limited to HIV/AIDS but would also be applied to other future crises [54]. Furthermore, during the COVID-19 crisis, spontaneous initiatives led by indigenous Maori effectively addressed the challenges in Aotearoa, New Zealand [55]. These initiatives provided health services, engaging the community and leaders to support people in need, and an effective organisation based on knowledge and linkages of the community was thus established and functioned effectively during the crisis [55]. Additional evidence from Fiji shows how the identification of leaders and key stakeholders who formed part of national task forces responding to the COVID-19 pandemic was also instrumental for isolation and assistance to the most vulnerable [47]. This evidence sheds light on the fact that, often, communities can self-organise and try to fill gaps left by government authorities on how to address crises, including operationalising equitable solutions. The literature in this area has already confirmed the crucial role played by island communities [56,57]. However, further research is required to uncover the

mechanisms that would facilitate and combine government and community actions when facing threats from external shocks.

Active participation and coordination among community actors can also be central in the provision of basic needs during crises. This appears to be even more important in the context of small and remote communities, where issues such as food supply and access to medicines can be a real threat, as was the case in the island of Noepe [58]. Enhanced communication and partnership among internal and external stakeholders and network interaction, including government authorities, NGOs, and the private sector, can ensure that all actors can optimally leverage all resources available. Especially during crises, such interactions may be useful for different types of emergencies and disruptions to also understand that some solutions that may seem inefficient on the surface (e.g., home-grown agriculture) but offer societal and well-being benefits that extend beyond mere market logic [58]. Along the same lines, many communities located in different Pacific island countries have contributed to a prompt adaptation to the lockdown measures, for example, by supporting home gardening and focusing on domestic food production to create a buffer for further shocks [59]. Similarly, in response to the harsh restriction of the lockdowns in the Pacific island of Guam, especially for long-term patients, a program that was designed to provide improved access to social support and specialized community services for persons with dementia and their family caregivers showed a strong community response, with approximately 50–60 family caregivers and persons with dementia participating in the program monthly [60].

Academia and community leaders can also interact and cooperate together to address existing vulnerabilities prior to and during crises, with the aim of informing policy at the local level. In the small island state of Malta, the fast-growing urbanisation rates and challenges of maintaining functioning infrastructure in light of crises create dynamics that can increase the exposure to self-inflicted vulnerabilities, which may be kept in check through identification exercises performed by academia, civil society organisations and wider societal actors, providing a policy with further assessments that may be used in suitable responses during crises [61]. Similarly, during the COVID-19 pandemic, many community leaders in Micronesia have actively advocated for data to be presented in a disaggregated manner so as to understand the real impacts and understand how to better assist Pacific island communities, thus resulting in more effective and tailor-made responses to the crisis [45]. Looking at the combined risks of COVID-19 and cyclones in the Atlantic region, community involvement can be crucial to develop steps that include staying informed about the latest weather and COVID-19 updates, having a plan for evacuation or sheltering in place, preparing an emergency kit with essential supplies, and following recommended safety guidelines, such as wearing masks, performing physical distancing and maintaining good hygiene practices. [48].

A further element that makes community involvement essential in preparation for, during, and in the aftermath of crises, is the effective implementation of emergency plans. Effective crisis response policies are not only based on direct central government instructions but require a respectful understanding of community structures and traditions [62,63], and active engagement with communities, including local governments. This passes through devising policies that generally recognise: i. the importance of family (knowledge sharing, events, etc.); and ii. ways of life and realities of living in response to crises (access to services, use of infrastructure, etc.). Without this type of engagement, any plan or policy to respond to the crises can be jeopardised [64].

Community values and identities can contribute to stopping the spread of pandemics, as shown by the response in the Falkland/Malvinas islands [65]. Here, the self-organisation of community members was enabled through 'social control' and positive behaviour reinforcement, adhering to rules and norms of protecting the island from COVID-19 spread. In the case of the Falkland/Malvinas, the island community identities acted as a glue and created virtuous behaviour, reducing the risks coming from the spread of the COVID-19. Doubts remain, however, on the capacity of the community to preserve such responses

given the changes in demographics due to immigration and the new dynamics that could occur [65].

The assessment of the literature gathered in this specific thematic area supports the existing body of knowledge on the key role played by local communities in the context of islands [23], possibly extending its reach toward understanding how instrumental local communities can be in responding to crises. This is especially the case when complementing the work of public authorities using the inherent resources of non-state actors and other resources stemming from social capital [52].

### 4.3. Local Government Support

Local authorities and governments can be key players in the response to climate and public health crises, especially for their physical vicinity to many households and the possible role that they can play in acting as a 'filter' between central government and local actors.

Local governments can support an optimal implementation of emergency plans, given their knowledge of the territory [58], or provide crucial information to build suitable infrastructure meaningful for communities prior to or after extreme climate events, especially in remote islands [66]. Effective leadership, especially by local public authorities, can play a key role in ensuring that communities follow emergency plans. A clear example of this is seen in the Falkland/Malvinas islands, where the local police devised solutions that balanced personal needs with rule compliance during the COVID-19 lockdowns [65]. Community health workers in Hawaii, especially during the COVID-19 pandemic, helped with the adaptation of emergency plans that might have overshadowed chronic long-term issues of part of the already vulnerable population [67].

### 4.4. Long-Term Strategic Planning

A common pattern that emerges from the literature reviewed is the need for government authorities in small islands to invest more in the long-term strategic planning and management of crises and to draw from the already relatively rich experience of past crises, especially by involving community members, local governments and the private sector in the design and implementation of such long-term planning.

Looking at the wider impacts that a crisis like COVID-19 has had on the population in many small islands, the necessary long-term planning identified in the Caribbean region looked at broad topics. The literature referred to topics such as mental health and well-being, and highlighted that the conditions for building social and health infrastructure capable of jointly confronting future crises exist [68]. Similarly, the assessments of ecological restoration projects (forests, mangrove, sea-meadows, etc.) usually consider only the benefits to the ecosystem, rarely including the impacts that such projects can have on the overall well-being of society, which were clearly evident during periods of lockdowns such as those experienced during the COVID-19 pandemic. Therefore, by having such projects associated with a wider group of stakeholders and considering a more holistic approach, they could bring long-term benefits for the whole society [69]. This interconnectedness of crises and their compound and stratified impacts are very often tackled separately when a holistic framework like the planetary health approach would provide a more comprehensive tool to tackle these crises. Integrating those positive short-term responses and transposing them to more long-term strategies is also another important element that emerges from evidence in SIDSs. In this regard, the necessity of facilitating remote work environments as a result of lockdowns can be beneficial for many SISs during normal periods of non-crisis as well; for instance, distant learning education, telemedicine, energy, and financial services, to mention a few, can still be provided remotely, especially in a context where local expertise is difficult to foster due to size [46] and inherent market competition challenges [70]. The long-term adoption of good practices that were developed during the COVID-19 crisis, such as online learning, training and telemedicine, can mitigate some of the disadvantages of remoteness, especially when double insularity occurs, such as

in the case of Trinidad and Tobago [71]. Similarly, in the Solomon Islands, online procedures can complement physical ones to assess the overall well-being of medical staff and can also monitor long-term performance and work satisfaction [47].

One of the most significant findings is the lack of an overarching framework in the peer-reviewed literature that allows for the addressing of climate and health crises together in a comprehensive manner. During crises, the disruption of basic needs supply chains are usually managed through traditional efficiency-driven and risk management supply chain management tools [58]. However, the proposition of tailor-made solutions to multiple crises around the disaster event cycle of i. preparation; ii. absorption; iii. recovery; and iv. adaptation would allow for a more systematic approach that could provide more sustainable responses, especially by actively involving all community actors. This 'Resilience-by-Design and Resilience-by-Intervention' seems to be better suited for capturing small islands' inherent characteristics and ensuring that basic services are also provided beyond basic market logics [58]. This goes hand in hand with providing continuous training for the sustainable use of land and agricultural practices to a wider portion of the population, as suggested for the Pacific islands countries, which can make use of idle land, increase preparedness in case of extreme climate events and connectivity restrictions due to pandemics, especially by actively involving women and youth [59]. Although successful responses to these crises often exhibit similar underlying factors, a further finding from the research results is that there is little coordination at the conceptual level on the approach and responses or in terms of preventively addressing the drivers of those simultaneous crises.

Long-term financial stability is also essential for planning risk management and increase preparedness to future crises. The debt-for-climate swaps represent a concrete possibility for many SIDSs who are highly indebted, specifically by using the debt cancellation, suspension or rescheduling into climate and health risk reduction strategies, to address their existing vulnerabilities [72].

## 5. Conclusions

The systematic review presented in this paper looked into the existing body of knowledge conducting research on climate resilience and health preparedness as a response to the climate and public health crises in the context of small islands. It found that in the peer-reviewed literature, there is not an overarching framework that allows for the addressing of climate and health crises together in a comprehensive manner. While efforts to tackle these crises simultaneously can contribute to the achievement of the SDGs, particularly SDG 13 focusing on taking urgent actions to combat climate change and its impacts (targets 13.1/13.3) and SDG 3 ensuring healthy lives and promoting well-being for all (targets 3.8/3.d), the findings show a lack of cooperation among stakeholders in small island settings. Although successful responses to these crises often exhibit similar underlying factors, there is little coordination at the conceptual level on how to approach responses or preventively address the drivers of those crises simultaneously.

However, specific actions to strengthen resilience were documented to have been effective when facing climate and health crises, which we categorised as: i. the management of short-term risks; ii. community action; iii. local government support; and iv. the role of long-term strategic planning.

In assessing the limitations of our research, several factors ought to be considered. Firstly, by focusing on the literature post-2007, we may have omitted some historical perspectives and theories, though we believe that the recent 15-year span of sources may have adequately made reference to key theories and concepts. Secondly, our study is confined to pandemics and environmental/climate crises, excluding other types of crises that could offer additional insights. Future research could broaden this scope. Thirdly, our systematic review approach did not encompass grey literature and non-peer-reviewed articles, presenting a potential avenue for more comprehensive future research. Lastly, while our focus on small islands provides specific insights, a comparison with larger

states was beyond the scope of this study, which might limit the generalisation of our findings. The diverse nature of islands—varying in topography, development levels and governance levels—further implies that our results and the recommended solutions to crisis preparedness may not be universally applicable.

Our analysis identifies four macro-factors contributing to resilience in small islands; however, assessing the effectiveness of the specific measures included in the papers remains a complex challenge due to the varied nature of the case studies, and there is a need to adopt a different methodological approach to perform this assessment. We acknowledge that future research could aim to develop methodologies for a more precise measurement of these resilience-building strategies, assessing their effectiveness across diverse island contexts perhaps by also using a planetary health approach. This would enhance a broader understanding of how these measures can be tailored and effectively implemented in specific island settings.

The key results of this systematic literature review also reveal that there is a relative scarcity of studies that address public health and climate crises together in the context of small islands. This is particularly relevant considering that the majority of studies concentrate on larger, non-insular states. A further element to ponder about is the fact that for those peer-reviewed papers that exist in the context of islands, there is a tendency to focus on the Pacific and Caribbean regions and significantly less on the Atlantic and Indian Ocean.

An additional point arising from the results is that there is room for coordinating the planning, designing and implementation of preparedness for climate and health crises in a joint manner, taking advantage of the use of generally scarce resources in islands. This alignment with the objectives set by the aforementioned SDGs not only advocates for a comprehensive approach to multidimensional crises but also underscores the wider commitment to sustainable development and holistic well-being. Finally, involving local communities, civil society organisations and local governments in both short-term response and long-term strategic planning can improve the resilience of governments and societies to crises. This calls for further research on the response and assessment of preparedness in the context of multiple crises in islands.

In examining the limited application of a planetary health approach in small islands, several key factors emerge. Firstly, there is often a lack of awareness or understanding of the interconnectedness between environmental and health systems, particularly in regions where immediate crises overshadow long-term planning, as in the cases pointed out in this study.

Additionally, small island states frequently face resource constraints, limiting their capacity to implement comprehensive, multidisciplinary strategies. The planetary health approach, while holistic and potentially effective, demands significant interdisciplinary collaboration, which can be challenging in contexts with limited expertise and infrastructure. Moreover, the political and economic focus in these regions is often on immediate survival and recovery rather than on integrated, long-term strategies. This short-term focus may inadvertently sideline holistic frameworks like planetary health, which require a shift in both mindset and resource allocation.

Therefore, while the planetary health approach holds promise, its integration into policy and practice in small island contexts necessitates overcoming these systemic barriers.

**Supplementary Materials:** The following supporting information can be downloaded at: https://www.mdpi.com/article/10.3390/su16020550/s1.

**Author Contributions:** Conceptualisation, S.M. and L.N.; methodology, S.M. and L.N.; software, S.M. and L.N.; validation S.M. and L.N.; formal analysis, S.M. and L.N.; investigation, S.M. and L.N.; resources, S.M. and L.N.; data curation, S.M. and L.N.; writing—original draft preparation, S.M. and L.N.; writing—review and editing, S.M. and L.N.; visualisation, S.M. and L.N.; supervision, S.M.; project administration, L.N.; funding acquisition, S.M. and L.N. All authors have read and agreed to the published version of the manuscript.

**Funding:** This research was funded by the University of Malta's Internal Research Grants Programme with the Research Excellence Award in December 2021 (I21LU13).

**Institutional Review Board Statement:** This study was conducted in accordance with the Declaration of Helsinki, followed the University of Malta research ethics and data protection procedures, and was approved by the Faculty of Economics, Management and Accountancy Ethics Committee (FEMA-2022-00411).

**Informed Consent Statement:** Not applicable.

**Data Availability Statement:** The data presented in this study are available on request from the corresponding author.

**Acknowledgments:** The authors would like to thank the Islands and Small States Institute chair, board and staff, Project Support Office, Knowledge Transfer Office, Finance Office and the pro-rector for Research and Knowledge Transfer from the University of Malta for the excellent collaboration to make this project a success.

**Conflicts of Interest:** The authors declare no conflicts of interest.

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
