# Peer review of "Improving Sustainability, Climate Resilience and Pandemic Preparedness in Small Islands: A Systematic Literature Review"

_sustainability, doi:10.3390/su16020550_

Round 1
Reviewer 1 Report
Comments and Suggestions for Authors
The paper presents a systematic review investigating the potential link between climate change and COVID-19, or whether any connection exists between the two. However, the paper lacks clear motivation. The authors should provide information addressing the following points:
a) What prompted the authors to explore the relationship between COVID-19 and climate change? While both are global crises, the motivation for examining their connection needs to be explicitly stated.
b) The authors are advised to verify the correctness of keywords such as "environm," "planeta," and "islan" used in the review. They should explain how these query strategies were developed. Utilizing tools like VOS Viewer could enhance the precision of the information retrieved.
c) The conclusions drawn in the paper are considered inadequate and should be enhanced with a more scientific approach. While concluding that there is no relationship between the two crises is a valid assertion, the authors should explore whether they discovered any other significant findings.
A major revision is recommended to address these issues.
Author Response
Thank you for providing your review.
Please see attachment.

Reviewer 2 Report
Comments and Suggestions for Authors
Dear Authors,
thank you for quite simple but very interesting and relevant research. I truly enjoyed reading your paper. However, I have a few minor suggestions to improve your paper.
1. Why do you use term "ii. non-governmental actors;" in the abstract while everywhere else there is "community actions"? Is there any justification for this or were you just looking for synonyms? I truly believe that term "community actions" is so much better revealed in your paper and it is so much more in the essence of community than NGO.
2. 118-119 lines. Why these databases? It would be nice to give at least a few sentences on justification for choosing these databases.
3. Please provide the limitations of your research in the Conclusions part.
4. Also, I would suggest elaborating on the future research to the greater detail in the Conclusion section.
Comments on the Quality of English LanguageMinor editing of English language required.
Author Response
Many thanks for providing your review and for the useful comments.
Please see the attachment.

Reviewer 3 Report
Comments and Suggestions for Authors
The subject of this article is very interesting. Here's my suggestion:
1. The authors put the most exciting part of the article in the discussion part. This is obviously unreasonable. I suggest moving the discussion part to the results part. And provide more research evidence.
2. I suggest that the authors highlight the characteristics of small islands. It would be good to explain the difference between small islands and other studies.
3.COVID-19 broke out in 2020. I suggest that authors differentiate between different time periods in their statistics. It would be good to be able to compare studies on COVID-19 with other studies.
Author Response
Many thank for providing your review and for the useful comments.
Please see the attachment.

Reviewer 4 Report
Comments and Suggestions for Authors
Content
----------
The paper is centered around small islands and focuses on the issues of climate change and health crises. It discusses the results of a systematic literature review, and then using the data collected from the official database to analyse the current state of knowledge on multidimensional crises in health and environmental dimensions. According to the results obtained from the analysis of the topic of the paper, it will be divided into four categories to discuss specific actions to strengthen resilience building in small islands.
Major comments
--------------
1. Topic
Is there a better word than “improving”, and it's not an exact word that describes what you do in this paper.
2. Logic Problems
The whole article has a big problem of logic and content connection of the full article. For example:
Line 43: What is the purpose of this paragraph, the logic is not smooth.
Line 459: What’s the main function of this paragraph?
3. Line 12:
“setting out to examine 12 which resilience-building measures are adopted ...”, you didn’t figure out this question in the main article.
4. Line 17:
“Within these, we examined whether a planetary-health approach, which recognizes the links between environment and health systems, was adopted”, I didn't find anything relevant in the main article.
5. Line 112:
This part is the center of the full article and the full article is to solve these two problems, it is suggested to write these two questions in the abstract.
6. Line 194:
Please provide some theoretical explanation and basis for the way of division.
7. Line 382:
Content mismatch. “ Communities’ adherence to emergency plans can ...”should be placed at 4.2.
Minor comments
--------------
8. Line 15/449:
According to Sustainable Development Goals and targets, The serial number is “3.d”.
9. Line 22:
Change to “community actions”
10. Figure 1:
Note that the chart remarks are case sensitive
11. Line 367:
Formatting pay attention to indentation.
Evaluation
--------------
Given the above, I'm in a position to major revision, and rewrite the abstract.
Comments on the Quality of English Language
Extensive editing of English language required
Author Response
Many thanks for your review and for providing useful comments.
Please see the attachment

Reviewer 5 Report
Comments and Suggestions for Authors
The article "Improving sustainability, climate resilience, and pandemic preparedness in small islands: A systematic literature review" explores the resilience-building measures adopted by small islands facing the simultaneous challenges of climate change and the COVID-19 pandemic. While the research addresses a relevant and pressing topic, there are several aspects that merit consideration:
Scope and Selection Criteria:
The article acknowledges that only 662 out of 16,369 studies mentioned small islands in the context of pandemic, health, and climate assessment. A discussion on the criteria for selecting these studies and potential biases introduced by such selection would strengthen the transparency and reliability of the review.
Main hook:
Authors must have to include the main hook of the study in the revised article.
Please include the given statement as the first sentence of the introduction with the given studies [1-2] as “Excessive burning of fossil fuels in production systems caused climate change and external shocks [1-4]. In particular, small islands….”
[1] https://doi.org/10.1016/j.apenergy.2021.118459
[2] https://doi.10.3389/fenvs.2023.1186328
Planetary Health Approach:
The article suggests that a planetary-health approach, recognizing the links between environment and health systems, is scarcely considered in small islands facing dual crises. It would be beneficial to explore why this approach is underutilized and discuss potential barriers or challenges in implementing such a holistic framework.
Effectiveness of Resilience Measures:
The article categorizes effective resilience measures into four groups: short-term risk management, non-governmental actors, local government support, and long-term strategic planning. A more in-depth analysis of the effectiveness of specific measures within each category and their applicability to different small island contexts would enhance the practical implications of the findings.
Data Quality and Rigor:
The review relies on the outcomes of the PRISMA protocol, but the article should provide more information on the quality assessment of the studies included. It is crucial to ensure that the selected studies meet rigorous standards to enhance the credibility of the overall findings.
Generalizability and Contextualization:
Small islands vary significantly in terms of geographical, socio-economic, and political contexts. The article could benefit from discussing the generalizability of findings across different types of small islands and providing insights into how specific contextual factors might influence the effectiveness of resilience measures.
Author Response
Many thanks for your review and for providing your useful comments.
Please see the attachment

Round 2
Reviewer 1 Report
Comments and Suggestions for Authors
The reviewer recommended incorporating VOS Viewer, and the authors have sufficiently explained its methodology, making it acceptable.
They adequately responded to this reviewer by providing substantial information. Based on this, I recommend accepting the paper.
Reviewer 4 Report
Comments and Suggestions for Authors
The author addressed all my previous concerns.
Comments on the Quality of English LanguageMinor editing of English language required